# EFFICIENT IDENTIFICATION OF DIRECT CAUSAL PARENTS VIA INVARIANCE AND MINIMUM ERROR TESTING

## ABSTRACT

Invariant causal prediction (ICP) is a popular technique for finding direct causes (causal parents) of a target via exploiting distribution shifts. Despite its targeted search, ICP still needs to run an exponential number of tests, which significantly limits its applicability, particularly in tasks with a large number of variables to consider. Furthermore, as others have pointed out, ICP fails to identify causes when distribution shifts only affect a few variables. We propose two approaches, MMSE-ICP and fastICP, which employ an error inequality to address the identifiability problem of ICP. The inequality states that the minimum prediction error of the predictor using causal parents is the smallest among all predictors which do not use descendants. fastICP is an efficient approximation that exploits the error inequality and a heuristic to reduce the number of tests for invariance required. Our experiments on simulated and real data show MMSE-ICP and fastICP outperforming competitive baseline approaches and fastICP being much more scalable.

## 1 INTRODUCTION

Causal discovery (CD) can offer insights into systems' behaviors (Pearl, 2018). These insights are helpful for crafting interventions to alter some outcomes (e.g. diseases) or for creating robust ML models. For example, a model based on a target's causal parents can robustly predict the target despite various changes (interventions). CD can be global or local: global CD searches for the complete causal graph while local CD only searches for causal relations surrounding a specific target $Y$. Thus, global CD is often intractable as the search domain grows exponentially with the number of variables. Local CD is more tractable and is sufficient if the goal is to change $Y$ through interventions or to build domain-invariant ML models of $Y$. Recently, many works have built on local CD to improve ML models' out-of-distribution generalization (Rojas-Carulla et al., 2018; Magliacane et al., 2018; Arjovsky et al., 2019; Christiansen et al., 2021).

Invariant causal prediction (ICP) is a local CD method that finds the causal parents of $Y$ using the invariance property: the distribution of $Y$ conditioned on all of its causal parents will be invariant under interventions on variables other than $Y$ (Peters et al., 2016). Specifically, ICP tests for invariance for all sets of variables and outputs the intersection (denoted as $\hat{\mathbf{S}}_{\text{ICP}}$) of all invariant sets. As ICP still needs to run an exponential number of tests for invariance, applying ICP to problems with numerous variables is difficult. Besides, ICP can only identify all causal parents when the number of interventions is at least equal to the number of variables (Peters et al., 2016). When this number is limited, $\hat{\mathbf{S}}_{\text{ICP}}$ may fail to identify causal parents (Rosenfeld et al., 2021; Mogensen et al., 2022), as disjoint sets of predictors can be invariant, yielding an empty intersection (see Figure 1). While some methods can find more accurate predictors than ICP can (Subbaswamy et al., 2019), these predictors could be less robust when only a few interventions are observed in training data. Besides, these methods are less suitable for crafting interventions to alter the target $Y$ because their outputs may include descendants of $Y$.

Invariant ancestry search (IAS) addresses ICP's failure to identify causal parents by outputting the causal ancestors of $Y$ instead (Mogensen et al., 2022). The causal ancestors identified is the union (denoted as $\hat{\mathbf{S}}_{\text{IAS}}$) of all minimally invariant sets. A set of predictors is minimally invariant if none of its proper subsets is invariant (Mogensen et al., 2022). Like ICP, $\hat{\mathbf{S}}_{\text{IAS}}$ can also be used to construct a stable predictor of $Y$ across environments even though it is not the most compact stable predictor. Besides, intervening on an ancestor may not be as effective in influencing the value of $Y$ as

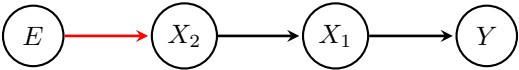

Figure 1: Consider this causal directed acyclic graph (DAG) as a motivating example. Let $E$ denote the variable capturing the environment (or context) (Mooij et al., 2020). Its children are directly affected by intervention. This reflect distribution shifts between contexts. $Y$ is the target variable. $\hat{\mathbf{S}}_{\text{ICP}} = \emptyset$ because the invariant sets are $\{X_1\}$, $\{X_2\}$, and $\{X_1, X_2\}$. In contrast, $\hat{\mathbf{S}}_{\text{IAS}} = \{X_1, X_2\}$. Our proposed methods output $\{X_1\}$ since the prediction error of $\hat{Y}_M(X_1)$ is less than $\hat{Y}_M(X_2)$'s.

intervening on a parent. IAS can be sped up by testing for invariance for sets of predictors up to a certain size at the cost of potentially finding fewer causal ancestors, thus trading accuracy for speed.

We propose two approaches that employ an error inequality to address the identifiability and scalability problem of ICP. The first is MMSE-ICP (short for *minimum mean squared error ICP*), which outputs the invariant set of variables with the smallest error for predicting $Y$. This is because, as we show below, the prediction error of the predictor using $Y$'s causal parents is the smallest among all predictors which do not use $Y$'s descendants. The second is fastICP, an efficient approximation that relies on a heuristic and constrained search scheme (see Section 3.4). fastICP also exploits the error inequality to reduce the number of tests for invariance to identify the causal parents. Consequently, fastICP can handle large-scale problems with a lot of variables to consider. Experiments on simulated and real data show MMSE-ICP and fastICP outperforming competitive baseline approaches.

## 2  RELATED WORK

There are many families of CD methods using solely observational data. Constraint-based (Spirtes et al., 2000) and score-based (Chickering, 2002) methods can identify up to the Markov equivalent class (MEC) of the true graph, leaving some edges with unresolved direction. With additional assumptions (e.g. non-Gaussianity (Shimizu et al., 2006), non-linearity (Hyvärinen & Pajunen, 1999; Hoyer et al., 2008; Zhang & Hyvärinen, 2009; Peters et al., 2014; Zhang et al., 2015), or independent causal mechanisms (Janzing et al., 2012)), methods are able to resolve more edges (Glymour et al., 2019). Recently, CD methods based on continuous optimization (Zheng et al., 2018; Geffner et al.) have become popular although they often have no identifiability guarantees (Mogensen et al., 2022).

Having data from multiple settings (observational and interventional) can improve identifiability. When the interventions are known, undirected edges in the MEC first estimated using observational data can then be resolved using interventional data (He & Geng, 2008). Other approaches jointly model interventional and observational data (Hauser & Bühlmann, 2012; 2015; Wang et al., 2017). There are also optimization-based methods that leverage both types of data (Lorch et al., 2021; Lippe et al., 2022). In contrast, UT-IGSP (Squires et al., 2020), SDI (Ke et al., 2019), and DCDI (Brouillard et al., 2020) are methods that can work with unknown interventions. ICP (Peters et al., 2016; Heinze-Deml et al., 2018; Pfister et al., 2019; Gamella & Heinze-Deml, 2020; Martinet et al., 2022) and IAS (Mogensen et al., 2022) both assume unknown interventions and only depends on the invariance property to identify causal structures. Our approaches also assume unknown interventions.

Since prototypical global CD methods such as PC (Spirtes et al., 2000; Colombo et al., 2014; Li et al., 2019) and GES (Chickering, 2002) can be slow, faster global CD methods (Cheng et al., 2002; Tsamardinos et al., 2006; Ramsey et al., 2017) have been proposed, although they make strong assumptions that may be inappropriate in some problems. In contrast, local CD methods only identify local causal structures surrounding a target variable. For example, several algorithms only search for the Markov Blanket (MB) of the target (i.e. its parents, children, and spouses) (Koller et al., 1996; Margaritis & Thrun, 1999; Tsamardinos et al., 2003; Gao & Ji, 2015; Yang et al., 2021). Consequently, local CD methods often have faster runtime or can afford to make milder assumptions leading to more accurate identification. ICP and IAS only search for parents and ancestors of the target respectively, omitting all descendants of the target. They both make very mild assumptions and offer guarantees in terms of false positive rates. Another local CD approach is CORTH (Soleymani et al., 2022) which is based on double ML (Chernozhukov et al., 2018). CORTH assumes that $Y$ is not a parent of any other variable (a strong assumption which can be unrealistic) and that $Y$ is a linear combination of other variables, and achieves linear runtime finding parents of $Y$.

## 3 METHOD

### 3.1 DEFINITIONS AND ASSUMPTIONS

We represent the causal relationship between all variables with a Directed Acyclic Graph (DAG), where each node is a variable and directed edges between nodes represent direct causal influence (Pearl, 2009). We use the usual notations from graphical models (Lauritzen, 1996). Specifically, $\mathsf{PA}(Y)$, $\mathsf{CH}(Y)$, $\mathsf{AN}(Y)$, $\mathsf{DE}(Y)$, and $\mathsf{ND}(Y)$ denote the parents, children, ancestors, descendants, and non-descendants of $Y$ respectively. Thus, $\mathsf{PA}(Y) \subseteq \mathsf{AN}(Y) \subseteq \mathsf{ND}(Y)$ and $\mathsf{CH}(Y) \subseteq \mathsf{DE}(Y)$. There is an additional node $E$ in the graph to denote the different environments (or contexts). A set of variables/predictors $\mathbf{S}$ is invariant if $Y \perp\!\!\!\perp E|\mathbf{S}$. As is common in the causal inference literature, we assume (1) no hidden confounder influencing the target variable $Y$, (2) no intervention on $Y$, (3) no feedback between variables, and (4) independence of mechanisms. Like prior work (Mogensen et al., 2022; Mooij et al., 2020), we also assume that (5) $E$ is exogenous (i.e., $E$ has no parent).

### 3.2 INVARIANT CAUSAL PREDICTION (ICP) AND INVARIANT ANCESTRY SEARCH (IAS)

ICP exploits distribution shifts between different environments $E$ to learn a subset of $\mathsf{PA}(Y)$ (Peters et al., 2016). Given a test for the hypothesis $H_{0,\mathbf{S}}$ that $\mathbf{S}$ is invariant, ICP outputs the intersection of all invariant subsets. Specifically, $\hat{\mathbf{S}}_{\mathrm{ICP}} := \bigcap_{\mathbf{S}:H_{0,\mathbf{S}} \text{ not rejected}} \mathbf{S}$. In contrast, IAS tries to identify a subset of $\mathsf{AN}(Y)$ (Mogensen et al., 2022). Let $\hat{\mathcal{I}}$ be the set of all sets $\mathbf{S}$ for which $H_{0,\mathbf{S}}$ is not rejected. Define $\widehat{\mathcal{MI}} := \left\{ \mathbf{S} \in \hat{\mathcal{I}} \mid \forall \mathbf{S}' \subsetneq \mathbf{S} : \mathbf{S}' \notin \hat{\mathcal{I}} \right\}$. Thus, $\hat{\mathbf{S}}_{\mathrm{IAS}} := \bigcup_{\mathbf{S} \in \widehat{\mathcal{MI}}} \mathbf{S}$. Although Peters et al. (2016) proposed two different tests for invariance, $H_{0,\mathbf{S}}$, the approximate test based on residuals of a linear predictor (Method II) is usually used because of its speed. Specifically, this statistical test checks whether the distribution (means and variances) of the linear predictor's errors across environments $E$ are the same. The predictor is fitted using the data from all environments.

### 3.3 IDENTIFYING CAUSAL PARENTS VIA MINIMIZING MEAN SQUARED ERROR

Instead of taking the intersection or union of the invariant predictors, we rely on the concept of the minimum mean squared error predictor to find the causal parents. The mean squared error (MSE) of a predictive function $\hat{Y}(\mathbf{X})$ with input variables $\mathbf{X}$ of target $Y$ is $\mathbb{E}_{\mathbf{X}Y}(Y - \hat{Y}(\mathbf{X}))^2$, where $\mathbb{E}$ denotes expectation. Let the minimum MSE achieved by $\hat{Y}_M(\mathbf{X})$ with input variables $\mathbf{X}$ be $\mathsf{MMSE}(\mathbf{X})$. Note that this is always with respect to predicting a target variable $Y$. For Figure 1's example, $\mathsf{MMSE}(\{X_2\}) \geq \mathsf{MMSE}(\{X_1\})$ so the causal parent $\{X_1\}$ forms the invariant predictor that minimizes MSE. Our MMSE-ICP algorithm generalizes the intuition from Figure 1. In the following, we provide some theoretical grounding for MMSE-ICP.

**Theorem 1** (Error Inequality). *Let $\mathbf{X}_1$ and $\mathbf{X}_2$ denote two sets of variables - not necessarily mutually exclusive. Then:* $\mathsf{MMSE}(\mathbf{X}_1 \cup \mathbf{X}_2) \leq \mathsf{MMSE}(\mathbf{X}_1)$. *Equality holds if* $Y \perp\!\!\!\perp \mathbf{X}_2|\mathbf{X}_1$.

*Proof.* By definition, $\mathsf{MMSE}(\mathbf{X}_1 \cup \mathbf{X}_2) = \mathbb{E}_{\mathbf{X}_1,\mathbf{X}_2,Y}(Y - \hat{Y}_M(\mathbf{X}_1,\mathbf{X}_2))^2$. Thus,

$$\mathsf{MMSE}(\mathbf{X}_1 \cup \mathbf{X}_2) = \mathbb{E}_{\mathbf{X}_1,\mathbf{X}_2,Y} \min_{\hat{Y}(\mathbf{X}_1,\mathbf{X}_2)} (Y - \hat{Y}(\mathbf{X}_1,\mathbf{X}_2))^2 \quad \text{(minimize over all predictors)}$$

$$= \mathbb{E}_{\mathbf{X}_1,\mathbf{X}_2} \min_{\hat{Y}(\mathbf{X}_1,\mathbf{X}_2)} \mathbb{E}_{Y|\mathbf{X}_1,\mathbf{X}_2}(Y - \hat{Y}(\mathbf{X}_1,\mathbf{X}_2))^2 \quad \text{(exchange order)}$$

$$\leq \mathbb{E}_{\mathbf{X}_1,\mathbf{X}_2} \min_{\hat{Y}(\mathbf{X}_1)} \mathbb{E}_{Y|\mathbf{X}_1,\mathbf{X}_2}(Y - \hat{Y}(\mathbf{X}_1))^2 \quad (\hat{Y}(\mathbf{X}_1) \text{ is in subspace of } \hat{Y}(\mathbf{X}_1,\mathbf{X}_2))$$

$$= \mathbb{E}_{\mathbf{X}_1} \min_{\hat{Y}(\mathbf{X}_1)} \mathbb{E}_{\mathbf{X}_2|\mathbf{X}_1} \mathbb{E}_{Y|\mathbf{X}_1,\mathbf{X}_2}(Y - \hat{Y}(\mathbf{X}_1))^2 \quad \text{(exchange order)}$$

$$= \mathbb{E}_{\mathbf{X}_1} \min_{\hat{Y}(\mathbf{X}_1)} \mathbb{E}_{Y|\mathbf{X}_1}(Y - \hat{Y}(\mathbf{X}_1))^2 = \mathbb{E}_{\mathbf{X}_1,Y} \min_{\hat{Y}(\mathbf{X}_1)} (Y - \hat{Y}(\mathbf{X}_1))^2 = \mathsf{MMSE}(\mathbf{X}_1)$$

If $Y \perp\!\!\!\perp \mathbf{X}_2|\mathbf{X}_1$, then $\mathsf{MMSE}(\mathbf{X}_1 \cup \mathbf{X}_2) = \mathsf{MMSE}(\mathbf{X}_1)$ because
$\min_{\hat{Y}(\mathbf{X}_1,\mathbf{X}_2)} \mathbb{E}_{Y|\mathbf{X}_1,\mathbf{X}_2}(Y - \hat{Y}(\mathbf{X}_1,\mathbf{X}_2))^2 = \min_{\hat{Y}(\mathbf{X}_1,\mathbf{X}_2)} \mathbb{E}_{Y|\mathbf{X}_1}(Y - \hat{Y}(\mathbf{X}_1))^2$. $\qquad\square$

**Corollary 1.1.** *For all subset $\mathbf{S}$ of $\mathsf{ND}(Y)$, $\mathsf{MMSE}(\mathbf{S}) \geq \mathsf{MMSE}(\mathsf{PA}(Y))$.*

---

**Algorithm 1:** MMSE-ICP

---

**Input:** $E, X_1, X_2, \ldots, X_d, Y$
**Output:** Potential parents of $Y$
1 **if** isInvariant$(Y, \emptyset)$ **then return** $\emptyset$
2 candidates $= []$
3 **for** *all* $\mathbf{S} \subseteq \{X_1, X_2, \ldots, X_d\}$ *in increasing cardinality order* **do**
4      **if** $\exists \mathbf{S}' \in$ candidates *such that* $\mathbf{S}' \subset \mathbf{S}$ **then** Skip line 5
5      **if** isInvariant$(Y, \mathbf{S})$ **then** candidates $=$ candidates $+ [\mathbf{S}]$
6 **return** $\arg\min_{\mathbf{S} \in \text{candidates}} \text{MMSE}(\mathbf{S})$

---

*Proof.* $Y \perp\!\!\!\perp \mathbf{S}|\text{PA}(Y)$ since $Y \perp\!\!\!\perp \text{ND}(Y)|\text{PA}(Y)$ (Causal Markov condition). This implies that $\text{MMSE}(\mathbf{S} \cup \text{PA}(Y)) = \text{MMSE}(\text{PA}(Y))$. Using Theorem 1, we have $\text{MMSE}(\text{PA}(Y)) = \text{MMSE}(\mathbf{S} \cup \text{PA}(Y)) \leq \text{MMSE}(\mathbf{S})$. $\qquad\square$

**Corollary 1.2.** *Let* $\mathbf{S}^* = \text{DE}(E) \cap \text{PA}(Y)$. *For any subset* $\mathbf{S}$ *of* $\text{DE}(E) \cap \text{ND}(Y)$, $\text{MMSE}(\mathbf{S}) \geq \text{MMSE}(\mathbf{S}^*)$.

*Proof.* Noting $Y \perp\!\!\!\perp \mathbf{S}|\mathbf{S}^*$, the proof follows that of Corollary 1.1. $\qquad\square$

**Theorem 2.** *If* $\mathbf{S}_1$ *is invariant, then* $\mathbf{S}_2 = \mathbf{S}_1 \cap (\text{DE}(E) \cap \text{ND}(Y))$ *is also invariant. In other words, removing all nodes in* $\mathbf{S}_1$ *that are not in* $\text{DE}(E) \cap \text{ND}(Y)$ *from* $\mathbf{S}_1$ *results in another invariant set.*

*Proof.* Since $\mathbf{S}_1$ is invariant, $Y \perp\!\!\!\perp E|\mathbf{S}_1$ so $\mathbf{S}_1$ blocks all paths between $E$ and $Y$, following d-separation (Pearl, 2009). All the paths between $E$ and $Y$ are out-going from $E$, as $E$ is exogenous (see Section 3.1). Thus, all nodes in $\mathbf{S}_1$ that are on paths between $E$ and $Y$ must also be in $\text{DE}(E)$.

Any node $X$ in $\mathbf{S}_1$ but not in $\text{DE}(E) \cap \text{ND}(Y)$ is either (1) on a blocking path between $E$ and $Y$ or (2) not on any blocking path. If it is the latter, then $\mathbf{S}_1 \setminus \{X\}$ still blocks all paths between $E$ and $Y$. If it is the former, then $X \in \text{DE}(E) \cap \text{DE}(Y)$. Since $X$ descends from both $E$ and $Y$, $X$ is on a blocking path first unblocked by collider(s). Removing $X$ together with all colliders will keep that path blocked. In addition, both $X$ and the colliders are in $\text{DE}(E) \cap \text{DE}(Y)$ so they are not in $\text{DE}(E) \cap \text{ND}(Y)$. In any case, invariance is not affected by excluding $X$.

Thus, excluding nodes in $\mathbf{S}_1$ that are not in $\text{DE}(E) \cap \text{ND}(Y)$ results in $\mathbf{S}_2$ that is also invariant. $\qquad\square$

**Proposition 1** (Identifiability). *Given that all tests of invariance return the correct results, Algorithm 1 will always find the subset of causal parents* $\mathbf{S}^* = \text{DE}(E) \cap \text{PA}(Y)$.

Proposition 1 follows from Corollary 1.2 and Theorem 2. Given invariant subsets of $\text{DE}(E) \cap \text{ND}(Y)$ containing $\mathbf{S}^*$, taking the subset with minimum MSE gives $\mathbf{S}^*$ according to Corollary 1.2 (Algorithm 1, line 6). Lines 3–5 create a list of invariant subsets of $\text{DE}(E) \cap \text{ND}(Y)$ containing $\mathbf{S}^*$.

Specifically, line 5 ensures that any subset added to the list is invariant. Lines 3–4 ensure that any set added to the list must be a subset of $\text{DE}(E) \cap \text{ND}(Y)$. Theorem 2 implies that for any invariant set $\mathbf{S}_1$ that is not a subset of $\text{DE}(E) \cap \text{ND}(Y)$, there exists an invariant set $\mathbf{S}_2$ that is a subset of $\mathbf{S}_1$ (having smaller cardinality than $\mathbf{S}_1$'s) and of $\text{DE}(E) \cap \text{ND}(Y)$. Since Algorithm 1 iterates through the sets of variables in increasing cardinality order (line 3), it will first find $\mathbf{S}_2$ and add $\mathbf{S}_2$ to the list. It will later exclude $\mathbf{S}_1$ because $\mathbf{S}_2$ is a subset of $\mathbf{S}_1$. Thus, all invariant sets that are not subsets of $\text{DE}(E) \cap \text{ND}(Y)$ will be excluded. $\mathbf{S}^*$ will always be in the list because no subset of $\mathbf{S}^*$ is invariant.

Proposition 1 indicates that the full set of causal parents of $Y$ is identifiable as long as $\text{PA}(Y) \subseteq \text{DE}(E)$. This implies that the number of interventions required can be much less than the number of variables in the graph. This condition is much more favorable than the condition specified in Peters et al. (2016) which requires interventions at every variable. Consequently, a single intervention at an upstream variable that affects all causal parents of $Y$ is sufficient for complete identification.

---

**Algorithm 2:** fastICP

---

**Input:** $E, X_1, X_2, \ldots, X_d, Y, \mathsf{MaxDepth}$
**Output:** Potential parents of $Y$

1 **if** isInvariant$(Y, \emptyset)$ **then return** $\emptyset$
2 $\mathbf{S} = \{X_1, \ldots, X_d\}$
3 **while** *not* isInvariant$(Y, \mathbf{S})$ **do**
4      candidates $= \{\mathbf{S}' \subseteq \mathbf{S} : |\mathbf{S}'| \geq |\mathbf{S}| - \mathsf{MaxDepth}\}$
5      **if** *no* candidates **then return** $\emptyset$
6      $\mathbf{S} = \arg\min_{\mathbf{S}' \in \text{candidates}} \mathsf{statDependency}(Y, \mathbf{S}')$
7 **while** *True* **do**
8      candidates $= []$
9      **for** *each* $Z \in \mathbf{S}$ **do**
10          $\mathbf{S}' = \mathbf{S} \setminus \{Z\}$
11          **if** isInvariant$(Y, \mathbf{S}')$ **then** candidates $=$ candidates $+ [\mathbf{S}']$
12      **if** *no* candidates **then break**
13      **else** $\mathbf{S} = \arg\min_{\mathbf{S}' \in \text{candidates}} \mathsf{MMSE}(\mathbf{S}')$
14 **return** $\mathbf{S}$

---

### 3.4 Faster search sequence

Though Algorithm 1 should be more accurate than ICP, it still requires an exponential number of tests for invariance. We develop a faster algorithm (Algorithm 2) which consists of two stages.

Stage 1 finds the largest invariant set of variables $\mathbf{S}$ (Algorithm 2, lines 2–6). Starting with the set of all variables, it removes potential colliders and their descendants, which might be unblocking paths from $E$ to $Y$, using the heuristic presented in Cheng et al. (1998). The heuristic assumes that blocking a path by removing some nodes from the conditioning set decreases the statistical dependency between $E$ and $Y$ (Cheng et al., 1998). In Algorithm 2, MaxDepth is a hyper-parameter for the maximum number of nodes to be removed at one time. The statistical dependency can be measured using the invariance test of Peters et al. (2016) where the lower the statistical dependency, the higher the probability of being invariant. Stage 2, based on Proposition 2, removes variables from $\mathbf{S}$ one-by-one while maintaining invariance (Algorithm 2, lines 7–13). By Proposition 2, when there is no invariant subset with smaller cardinality, $\mathbf{S} = \mathbf{S}^*$ and Algorithm 2 terminates.

**Proposition 2.** *Given that $\mathbf{S}$ is invariant, $\mathbf{S}^* \subseteq \mathbf{S}$, and $\mathbf{S}_{-Z} := \mathbf{S} \setminus \{Z\}$ are subsets of $\mathbf{S}$:*
*- If no subset $\mathbf{S}_{-Z}$ is invariant, then $\mathbf{S}^* = \mathbf{S}$.*
*- If there are invariant subsets $\mathbf{S}_{-Z}$, then $\mathbf{S}^* \subseteq \arg\min_{\mathbf{S}_{-Z}} \mathsf{MMSE}(\mathbf{S}_{-Z})$.*

*Proof.* If $\mathbf{S}$ is invariant but the subsets $\mathbf{S}_{-Z}$ are not, removing any node $Z$ from $\mathbf{S}$ unblocks a path from $E$ to $Y$. Thus, each node $Z$ in $\mathbf{S}$ lies on a blocking path. If there are multiple nodes on the same blocking path, one node (e.g. $W$) can be removed and $\mathbf{S}_{-W}$ is still invariant which is contradictory. Hence, each node $Z$ in $\mathbf{S}$ lies on a different blocking path. Since $\mathbf{S}^* \subseteq \mathbf{S}$, $Z \in \mathbf{S}^*$ or $\mathbf{S} = \mathbf{S}^*$.

Otherwise, let $\mathbf{S}_{-Z'} = \arg\min_{\mathbf{S}_{-Z}} \mathsf{MMSE}(\mathbf{S}_{-Z})$. Assume on the contrary that $\mathbf{S}^* \not\subseteq \mathbf{S}_{-Z'}$. However, $\mathbf{S}^* \subseteq \mathbf{S}$ so $Z'$ must be in $\mathbf{S}^*$. In addition, as $\mathbf{S}_{-Z'}$ is still invariant, there must be another node, denoted $T$, on the same blocking path as $Z'$ and $T$ is in $\mathbf{S}_{-Z'}$. Thus $\mathbf{S}_{-T} = \mathbf{S}_{-Z'} \cup \{Z'\} \setminus \{T\}$. By Theorem 1, $\mathsf{MMSE}(\mathbf{S}) \leq \mathsf{MMSE}(\mathbf{S}_{-Z'})$. Also by Theorem 1, $\mathsf{MMSE}(\mathbf{S}) = \mathsf{MMSE}(\mathbf{S}_{-T})$ since $Y \perp\!\!\!\perp \mathbf{S}|\mathbf{S}_{-T}$. Thus, $\mathsf{MMSE}(\mathbf{S}_{-T}) \leq \mathsf{MMSE}(\mathbf{S}_{-Z'})$ which is contradictory because $\mathbf{S}_{-Z'} = \arg\min_{\mathbf{S}_{-Z}} \mathsf{MMSE}(\mathbf{S}_{-Z})$. Consequently, $\mathbf{S}^* \subseteq \mathbf{S}_{-Z'}$. $\qquad\square$

### 3.5 Complexity Analysis

For a problem with $V$ variables, the complexity of ICP and MMSE-ICP is $O(2^V)$ as the number of invariance tests run is exponential. Stage 1 of fastICP is $O(V * 2^{\mathsf{MaxDepth}})$ because of the number of candidates checked grows exponentially with MaxDepth. Stage 2 is $O(V^2)$. Thus, the complexity of fastICP is $O(V * 2^{\mathsf{MaxDepth}} + V^2)$. When MaxDepth is less than $V$, fastICP is faster than ICP.

Table 1: Different settings with different number of nodes $(X_1, \ldots, X_d, Y, E)$, graph densities, number of interventions $(N_{int})$, and type of interventions. MB: Markov Blanket

| No. | No. of nodes ($d$+2) | Avg. density | Avg. MB size | $N_{int}$ | Type |
|---|---|---|---|---|---|
| 1. | 6+2 | 0.240 | 2.98 | 6 | Perfect |
| 2. | 6+2 | 0.145 | 2.58 | 1 | Perfect |
| 3. | 6+2 | 0.158 | 2.89 | 1 | Imperfect |
| 4. | 6+2 | 0.153 | 2.86 | 1 | Noise |
| 5. | 100+2 | 0.010 | 3.30 | 1–5 | Perfect |
| 6. | 100+2 | 0.050 | 26.99 | 1–5 | Perfect |

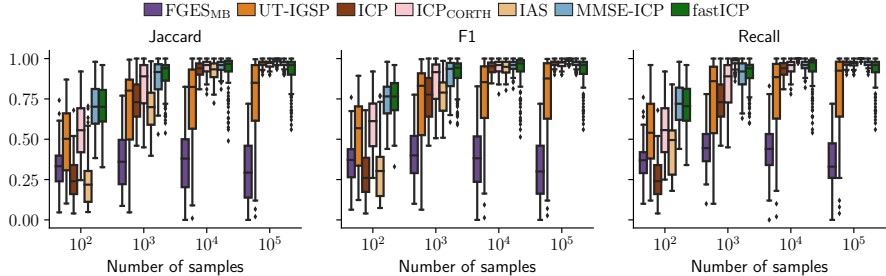

Figure 2: Performance when $N_{int} = d = 6$ (Table 1, No. 1). Reference set: PA$(Y)$.

## 4 SIMULATION EXPERIMENTS

### 4.1 DATA

We generate synthetic datasets using linear models with i.i.d. Gaussian noise. Similar to Peters et al. (2016); Mogensen et al. (2022), we consider 2-environment settings (one observational and one intervention). In each setting, 100 graphs are randomly generated. In a graph, beside the target node $Y$ and the environment indicator node $E$, there are $d$ additional nodes $\{X_1, \ldots, X_d\}$. The edges from $E$ to a subset of $\{X_1, \ldots, X_d\}$ specify the nodes that may be intervened on. The number of interventions ($N_{int}$) varies between different settings. For each graph, 50 sets of edge coefficients are drawn randomly. For each set of coefficients, we sample 4 datasets with different sample sizes ($10^2$, $10^3$, $10^4$, $10^5$). The data are standardized along the causal order to prevent shortcut learning (Reisach et al., 2021). In the interventional environment, interventions are applied to a random subset of children of $E$. We consider 3 types of interventions: (1) perfect intervention which severs all causal dependencies from parents, (2) imperfect intervention which modifies causal relationships between a node and its parents, and (3) noise intervention in which the intervened variable's noise variance changes (Cooper & Yoo, 1999; Tian & Pearl, 2001; Eberhardt & Scheines, 2007; Peters et al., 2016). The different settings are summarized in Table 1 (see Appendix A for more details).

### 4.2 BASELINES AND IMPLEMENTATION DETAILS

We compare MMSE-ICP and fastICP against ICP[1], IAS[2], fGES-MB[3], UT-IGSP[4], and ICP-CORTH. ICP-CORTH uses ICP to remove false positives in CORTH's output. To keep the runtime of the invariance test comparable to ICP and IAS, MMSE is estimated by averaging the prediction residuals. Since CORTH assumes $Y$ is a linear combination of its parents (similar to the simulation setup), ICP-CORTH is a competitive baseline. To test for invariance, MMSE-ICP and fastICP use the same test as ICP and IAS (Peters et al. (2016)'s Method II). Specifically, the test checks whether the mean and variance of the prediction residuals (of a linear regression model) is equal across environments.

---

[1] https://cran.r-project.org/package=InvariantCausalPrediction

[2] https://github.com/PhillipMogensen/InvariantAncestrySearch

[3] https://github.com/cmu-phil/tetrad

[4] https://github.com/uhlerlab/causaldag

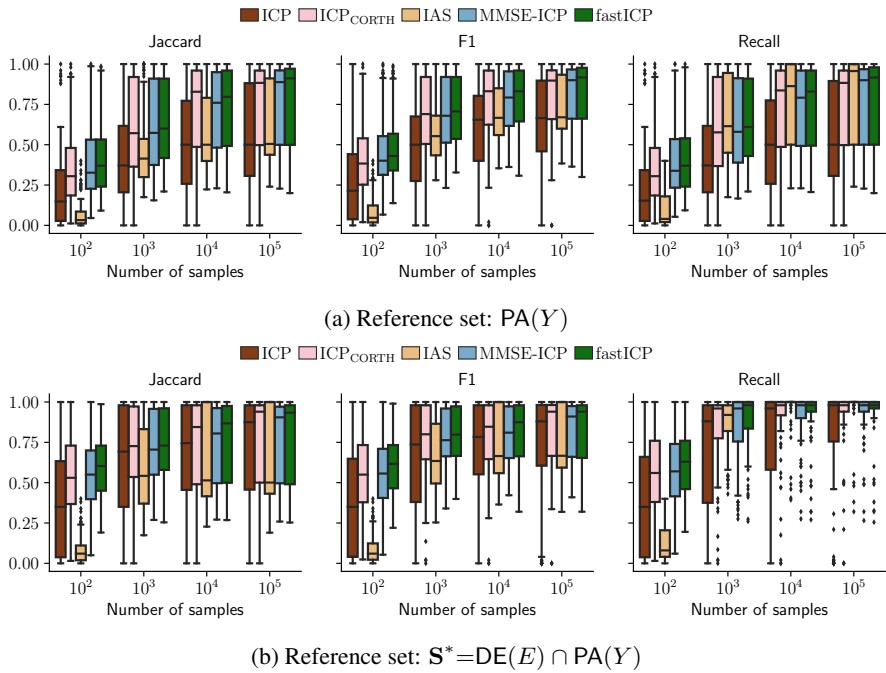

(a) Reference set: PA($Y$)

(b) Reference set: $\mathbf{S}^*=\text{DE}(E) \cap \text{PA}(Y)$

Figure 3: Performance when $N_{\text{int}}1; d = 6$ (Table 1, No. 2).

The MaxDepth hyper-parameter is set at 2. For ICP, IAS, UT-IGSP, MMSE-ICP, and fastICP, the significance level $\alpha$ is set at 0.05. The parameters of IAS ($C$ and $\alpha_0$) are set according to the original paper (Mogensen et al., 2022). Since it is intractable to search exhaustively using ICP, IAS and MMSE-ICP for large $d$, their search scopes are restricted in these cases. In particular, ICP and MMSE-ICP only search within an estimated MB of size 10. The 10 variables that are considered for further analysis are determined via L2-boosting (Friedman, 2001; Bühlmann & Yu, 2003; Hothorn et al., 2010). IAS only tests for sets with a single element.

*Jaccard similarity* and *F1-score* are used as metrics. The prediction is compared against a reference set, which could be either (1) the full set parents PA($Y$) and (2) the set of discoverable parents $\mathbf{S}^*$. Since IAS finds ancestors instead of parents, we also look at the parents' *Recall* rate.

## 4.3 RESULTS

When, $N_{\text{int}} = d = 6$, invariance-based algorithms should be able to discover all parents (i.e. $\mathbf{S}^* = \text{PA}(Y)$). Figure 2 shows their results in this setting. Their average runtimes are reported in Appendix C. With sufficient samples, invariance-based algorithms outperform fGES-MB (observational constraint-based CD) and UT-IGSP. When the number of interventions is limited (i.e. $N_{\text{int}} < d$, Table 1, No. 2–6), identifying all direct parents using invariance is more challenging since $\mathbf{S}^* < \text{PA}(Y)$. Due to the strict inequality, we only our approaches against invariance-based algorithms. For perfect interventions, MMSE-ICP and fastICP achieve similar performance and outperform the baselines in both Jaccard similarity and F1-score (Figure 3). The recall of our approaches is the same as IAS's and higher than ICP's. Figure 3b show that our approaches often found all the discoverable parents ($\mathbf{S}^*$). The same trends are observed for imperfect interventions (Figure 7 in Appendix), noise interventions (Figure 8 in Appendix), and large sparse graphs (Figure 4). They hold for large sparse graphs since the estimated MB of size 10 should cover the true MB most of the time (see Table 1).

For large dense graph (see Figure 5), both MMSE-ICP and fastICP still outperform the baseline approaches. Since the estimated MB is much smaller than the true MB, methods that cannot search exhaustively will miss out a lot of parents. ICP-CORTH performs worse than ICP probably because CORTH's assumption that the target Y has no children are more likely to be wrong. When the graph is large and dense, MMSE-ICP cannot search exhaustively so the ability to search through all nodes gives fastICP an edge over MMSE-ICP.

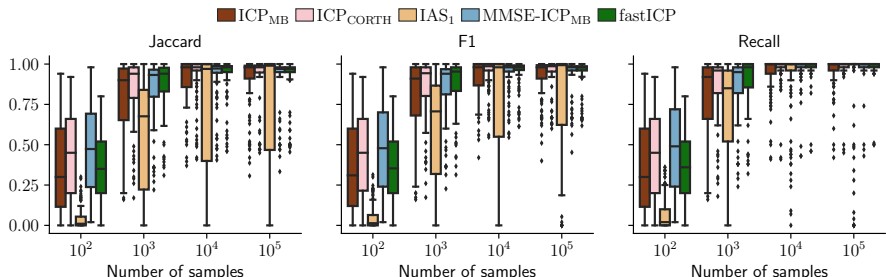

Figure 4: Performance for large sparse graphs (Table 1, No. 5). Reference set: $\mathbf{S}^{*}=\mathrm{DE}(E)\cap\mathrm{PA}(Y)$.

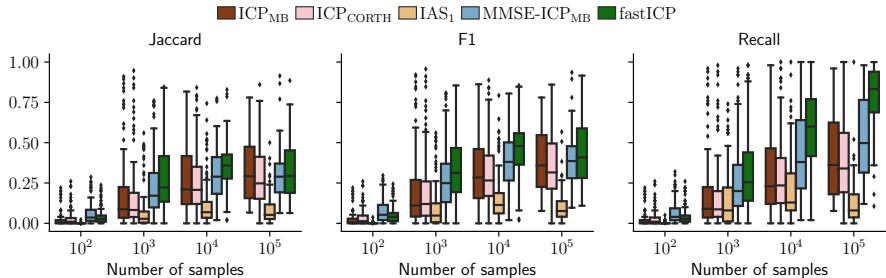

Figure 5: Performance for large dense graphs (Table 1, No. 6). Reference set: $\mathbf{S}^{*}=\mathrm{DE}(E)\cap\mathrm{PA}(Y)$.

## 5 EXPERIMENT ON GENE EXPRESSION DATA

### 5.1 DATA

We apply MMSE-ICP and fastICP to a real-world large-scale yeast gene expression dataset with 6170 genes (variables) (Kemmeren et al., 2014). There are 160 observational samples and 1479 interventional samples. Each interventional sample corresponds to an experiment where single gene $X_k \in \{1, \ldots, 6170\}$ has been deleted. Following Peters et al. (2016); Mogensen et al. (2022), we assume that a direct causal effect $X \rightarrow Y$ exists (true positive) if the expression level of gene $Y$ after intervening on gene $X$ lies in the 1% lower or upper tail of the observational distribution of gene $Y$.

Since neither the ground-truth graph or a separate set of validation data is available, we must use the same data for validation. Hence, we employ the same cross-validation scheme used in the original ICP paper to prevent information leakage in this experiment (Peters et al., 2016). Specifically, when predicting whether $X_k$ is a parent of $Y$, we do not include the sample when intervening on $X_k$ (if the sample exists) in the data used for inference. The interventional samples are split into 3 folds. In each fold, two thirds of the interventional samples not containing $X_k$ are used as interventional data, and remaining interventional data are used for validation. Thus, for each target, we need to run the the algorithms the same number of times s the number of folds. Additionally, when looking for potential causes of $Y$, we exclude samples corresponding to intervention on $Y$ (if it exists).

### 5.2 BASELINES AND IMPLEMENTATION DETAILS

ICP and IAS are used as baselines as they give confidence estimates for individual predicted parents/ancestors. The invariance test is the same as in the simulation experiments although the threshold is set at $\alpha = 0.01$, following Peters et al. (2016). L2-boosting (Friedman, 2001; Bühlmann & Yu, 2003; Hothorn et al., 2010) is used to estimate the MB of size 10 for ICP and MMSE-ICP. Even though fastICP can search exhaustively in simulations of 100 variables, it would be too slow for this problem. Hence, we restrict the fastICP search scope to an estimated MB of 100 variables. ICP-CORTH is excluded because it takes more than 3 days for one gene so obtaining the result for 6170 genes takes more than 18000 days. Instead scoring the methods' output at fixed thresholds, we ranked the set of predicted causal relations and score the most confident predicted relations.

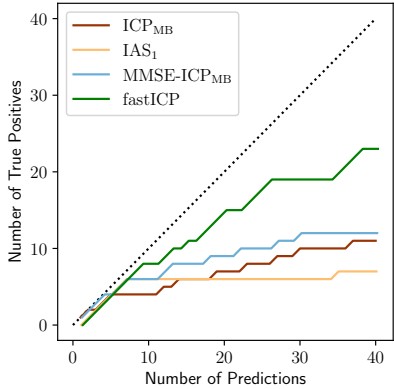

Figure 6: Performance on genes' parents prediction. The dotted line delineates perfect accuracy.

## 5.3 RESULTS

Figure 6 shows how the number of true positives vary as the methods are allowed to make more predictions. When the number of allowed predictions is lower than 8, the performance of the methods are very similar. However, when the number of predictions is more than 10, MMSE-ICP is generally more accurate than ICP. Besides, fastICP is the best approach for this task as it can afford to search more widely for the parents of each gene. The number of true positives for fastICP surpasses 20 when making 30 predictions (66% success probability).

## 6 DISCUSSION

While it is common to use only observational data for CD, interventional data with mechanism changes are very valuable for inferring the underlying causal model. In fact, changes (e.g. different equipments, locations, demographics, weathers) often arise naturally in realistic data collection. Given their abundance, harnessing these changes for CD would reveal insights useful for understanding and manipulating complex systems. However, exploiting changes in general is difficult as they are often imperfect interventions, merely altering the mechanism generating a variable. In contrast, controlled experiments are perfect interventions that fix designated variables to predetermined values (Tian & Pearl, 2001; Eberhardt & Scheines, 2007). Moreover, the variables that change are often unknown (unknown intervention targets). For example, gene knockout technologies are known to have off-target effects so the precise targets are unknown (Squires et al., 2020). As such, invariance-based CD methods are appealing because they can work with unknown perfect and imperfect interventions.

Like other invariance-based CD methods, ours also make no assumption about where interventions are and what precisely the effect of interventions may be. We only assume that there is no intervention on the target. This make our approaches appealing in many problems whereby specifying what an intervention or change of environment actually means is difficult. Besides, our work may have implications for representation learning methods that implicitly use CD (Arjovsky et al., 2019), since variables are not clearly defined so it is impossible to specify effects of interventions. In addition, unlike ICP which needs a sufficient number of interventions to identify all parents (Rosenfeld et al., 2021; Mogensen et al., 2022), our approaches can identify all parents even if they are only indirectly effected by interventions. Thus, our approaches would work very well with natural changes that effect several nodes at once (e.g. different hospitals with different equipments, doctors, and demographics).

In this work, we proposed two algorithms: MMSE-ICP which has similar runtime but better recall than ICP and fastICP which is more scalable than MMSE-ICP and more suitable for large-scale problems. Despite its speed, fastICP still has worst-case exponential complexity and whether there is a general polynomial-complexity algorithm remains an open question. Besides, as fastICP is a greedy algorithm, its performance depends on the robustness of the invariance test. Although fastICP used an approximate test based on linear regression for fair comparisons against baselines, we can adopt more complex non-linear tests (e.g. using neural nets) to increase detection sensitivity and robustness.

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
