# A    SIMULATION DETAILS

Different synthetic datasets are generated by varying the following parameters: (1) number of predictors $d$, (2) number of interventions $N_{\text{int}}$, and (3) the type of intervention. For each set of parameters, the following procedure is repeated 100 times to generate 100 different random graphs.

1. Sample a random acyclic graph $\mathcal{G}$ with $d+1$ nodes and a pair of nodes in $\mathcal{G}$ is connected with probability $p_{\text{edge}}$ (which is 0.1 for the large dense graph and is $2/N_{\text{int}}$ otherwise).
2. Choose a random node with at least 1 parent to be $Y$.
3. Add a node $E$ with no incoming edges. From of the set $X_1, \ldots, X_d$, pick $N_{\text{int}}$ nodes.
4. If $Y$ is not a descendant of $E$, repeat steps 1–3 until a graph where $Y \in \mathsf{DE}(E)$ is obtained.

$E$ is an environment indicator. A data sample is observational when $E = 0$ and is interventional when $E = 1$ (the children of $E$ may be intervened on).

For each graph, 50 sets of edge coefficients ($\beta_{i \to j}$) are drawn randomly. The coefficients are sampled independently and uniformly from the interval $U((-2, 0.5) \cup (0.5, 2))$. For each set of coefficients, we sample 4 datasets with different sample sizes $n \in \{10^2, 10^3, 10^4, 10^5\}$. Each data sample is generated as follow.

1. Sample $E$ from a Bernoulli distribution with probability $p = 0.5$.
2. Iterate through the nodes in graph in topological order and generate its value:
   - If $E = 1$ and the node is a child of $E$, the value depends on the type of intervention.
   - Else, the value is the sum of its parents' values weighted by the edge coefficients. Gaussian noise is added afterward.
   $$X_j = \sum_{i \in \mathsf{PA}(X_j)} \beta_{i \to j} X_i + N(0, 1)$$

We consider 3 types of interventions: (1) perfect interventions, (2) imperfect interventions, and (3) noise interventions (Cooper & Yoo, 1999; Tian & Pearl, 2001; Eberhardt & Scheines, 2007; Peters et al., 2016).

- Perfect interventions: the values of the intervened nodes are set to 1, regardless of the values of their parents.
- Imperfect interventions: the values of the intervened nodes are still the weighted sum of their parents. However, the edge coefficients are modulated by coefficients $\gamma_{i \to j} \sim U(0.0, 0.2)$.
$$X_j = \sum_{i \in \mathsf{PA}(X_j)} \gamma_{i \to j} \beta_{i \to j} X_i + N(0, 1)$$

- Noise interventions: the values of the intervened nodes are still the weighted sum of their parents. However, the additive noise is $N(0, 4)$ instead of $N(0, 1)$.

After all data samples are generated, the data are standardized along the causal order to prevent shortcut learning (Reisach et al., 2021).

# B    ADDITIONAL RESULTS

For noise interventions, MMSE-ICP and fastICP achieve similar performance and outperform the baselines in both Jaccard similarity and F1-score (Figure 8). The same trends are observed for imperfect interventions (Figure 7). Although MMSE-ICP and fastICP outperform ICP and IAS for imperfect interventions, there is still a large variance in the Jaccard similarity and F1-score of MMSE-ICP and fastICP. This could be because the approximate test based on residuals of a linear predictor does not have sufficient power to detect this types of changes in mechanisms. Switching to an invariance test with higher detection power might result in better results and lower variance.

ICP, MMSE-ICP, and fastICP obtain very similar results for large sparse graphs (Figure 9). However, for large dense graph (Figure 10), both MMSE-ICP and fastICP outperform the baseline approaches.

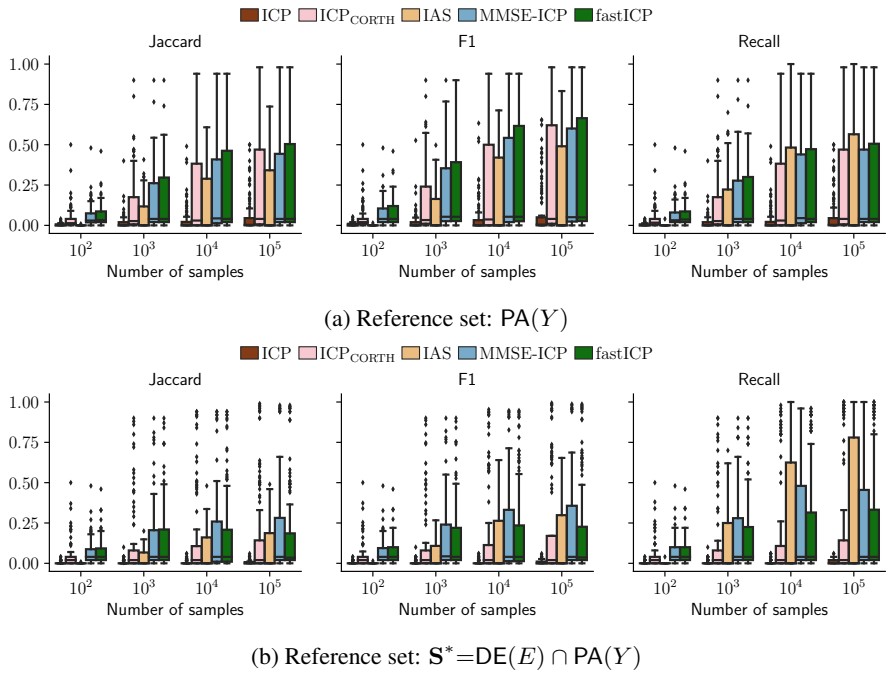

Figure 7: Performance under "imperfect" interventions when $d = 6$; $N_{\text{int}} = 1$ (Table 1, 3rd row).

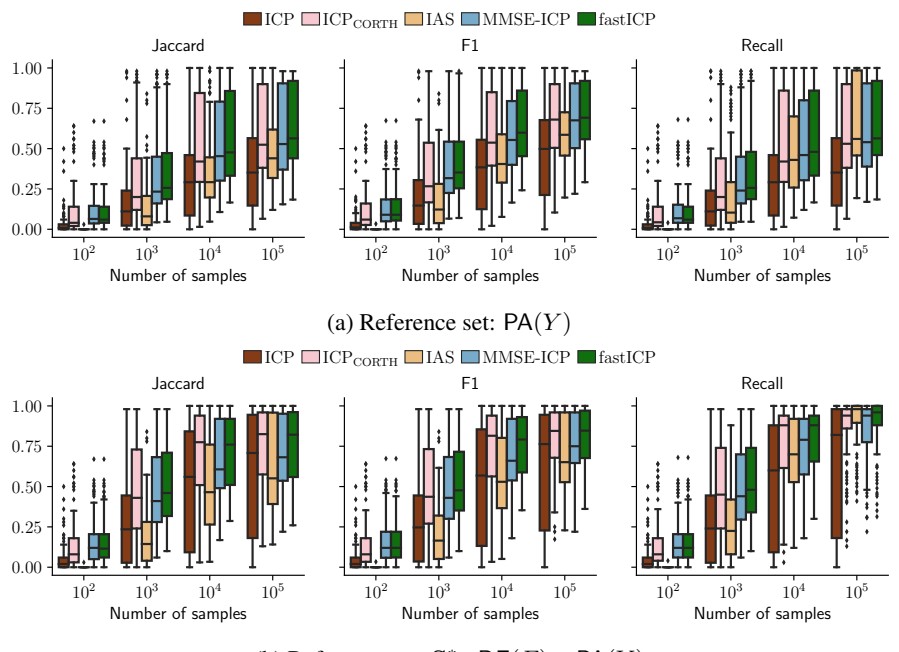

Figure 8: Performance under "noise" interventions when $d = 6$; $N_{\text{int}} = 1$ (Table 1, 4th row).

Although MMSE-ICP and fastICP achieve similar performance for small graph or large sparse graph in which MMSE-ICP can search exhaustively, when the graph is large and dense, the ability to search through all nodes gives fastICP an edge over MMSE-ICP.

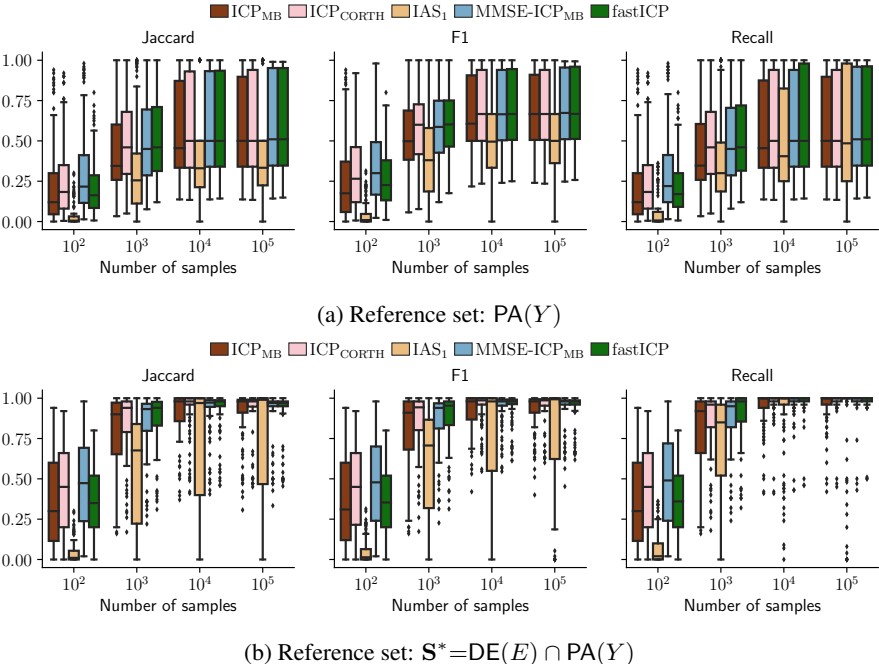

(a) Reference set: $\mathsf{PA}(Y)$

(b) Reference set: $\mathbf{S}^* = \mathsf{DE}(E) \cap \mathsf{PA}(Y)$

Figure 9: Performance for large sparse graphs (Table 1, No. 5).

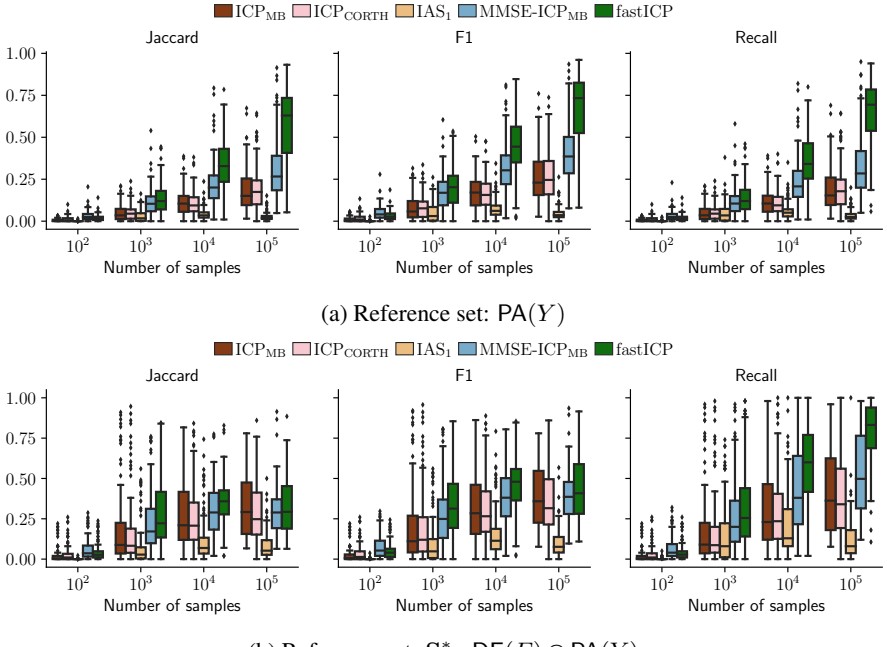

(a) Reference set: $\mathsf{PA}(Y)$

(b) Reference set: $\mathbf{S}^* = \mathsf{DE}(E) \cap \mathsf{PA}(Y)$

Figure 10: Performance for large dense graphs (Table 1, No. 6).

Table 2: The average runtime in seconds of different methods when $N_{int} = d = 6$ at various sample size $N$. Their accuracies are reported in Figure 2.

| Method | Language | $N = 10^2$ | $N = 10^3$ | $N = 10^4$ | $N = 10^5$ |
|---|---|---|---|---|---|
| FGES$_{MB}$ | Java | $2.342 \pm 0.028$ | $0.246 \pm 0.004$ | $0.364 \pm 0.006$ | $1.340 \pm 0.019$ |
| UT-IGSP | Python | $0.026 \pm 0.002$ | $0.040 \pm 0.008$ | $0.059 \pm 0.017$ | $0.607 \pm 0.223$ |
| ICP | R | $0.190 \pm 0.006$ | $0.325 \pm 0.004$ | $1.436 \pm 0.026$ | $20.296 \pm 0.421$ |
| ICP$_{CORTH}$ | R | $1.662 \pm 0.104$ | $1.857 \pm 0.122$ | $3.759 \pm 0.543$ | $28.312 \pm 7.061$ |
| IAS | Python | $0.016 \pm 0.007$ | $0.056 \pm 0.010$ | $0.112 \pm 0.022$ | $0.659 \pm 0.132$ |
| MMSE-ICP | Python | $0.035 \pm 0.010$ | $0.067 \pm 0.015$ | $0.138 \pm 0.027$ | $0.932 \pm 0.187$ |
| fastICP | Python | $0.028 \pm 0.002$ | $0.036 \pm 0.002$ | $0.070 \pm 0.004$ | $0.516 \pm 0.030$ |

## C  RUNTIME MEASUREMENT

We recorded the time each method took when the number of nodes ($d$) and the number of interventions ($N_{int}$) are equal to 6. This scenario is chosen because all the methods can complete exhaustive search for a problem of this size, thus allowing us to conduct a numerical comparison between all methods. For problems with 100 variables, ICP and IAS will not be able to search through all subsets. Table 2 reports the numbers of seconds elapsed when executing on an Intel Xeon 6126 CPU core (@ 2.60GHz). Since the official implementation of FGES$_{MB}$, ICP, and ICP$_{CORTH}$ are not written in Python so direct comparison with fastICP should be viewed with caution. However, we can see that

1. Among ICP-like methods with exponential worst-case complexity, fastICP is faster than MMSE-ICP, ICP, ICP$_{CORTH}$, and IAS.
2. fastICP is also slightly faster than UT-IGSP for large sample size (1E5 samples).