# OpenReview forum: "Efficient Identification of Direct Causal Parents via Invariance and Minimum Error Testing"
_ICLR.cc/2024/Conference — Submitted to ICLR 2024_

### Official Review · Reviewer_ZvvZ · 2023-10-29

**Soundness:** 3 good
**Presentation:** 3 good
**Contribution:** 2 fair
**Rating:** 6
**Confidence:** 3

**Summary:**

This paper focuses on identifying direct causes of a target variable under possible intervention environments by leveraging the invariant causal prediction (ICP) mechanism. By the basic observations that the prediction error for Y using its causal parents is more minimal than other predictors that contain non-parent variables, the authors present two approaches that learn the invariant set of variables with the smallest error for predicting Y, i.e., MMSE-ICP and fastICP, where the latter method has lower complexity. The experimental results, both on the simulated data and gene expression data, show the proposed methods are able to learn the correct ICP set with increasing samples.

**Strengths:**

1. The authors gave a good overview on the related literature.

2. This paper is well-written and well-organization.

3. The experimental results verify the efficientness of the proposed method of the proposed methods.

**Weaknesses:**

1. My main concern is how to obtain the minimized MSE in practical application, which is a key issue to the proposed theoretical results and it should be properly proofed if possible.

2. I note that the simulation data is only generated by a linear model. It should be clearly defined if the proposed theoretical results only focus on the linear model.

3. If the non-parents node has no noise term, e.g., X_2=2E and X_1=X_2, it seems that the MMSE(X_1 X_2) is equal to (X_1), which may not ensure the correct ICP set output by the proposed method. In other words, the author should claim the causal model used in this paper, such as SCM or ANM. If my understanding is not correct, can you clarify what generation process assumptions you are making?

**Questions:**

1. My question is how to test the constraint "isInvariant(Y,S)" (refer to the algorithm). Can you give some illustrations to its implementations?

---

> ### Author Response · Authors · 2023-11-14
>
> > My main concern is how to obtain the minimized MSE in practical application, which is a key issue to the proposed theoretical results and it should be properly proofed if possible.
>
> We acknowledge that defining an effective strategy to obtain the MSE is important to ensure the proposed methods return valid answers. One could employ more sophisticated methods such as cross-validation or doubly-robust machine learning to obtain more accurate MSE estimates. For example, CORTH (Soleymani et al. 2022) leverages doubly-robust machine learning to devise a more accurate test for causal parents. However, CORTH is also a more computationally costly procedure which makes applying ICP_CORTH to the gene expression problem not tractable. The MSE estimation scheme that we adopted seems to give useful results in practice and further exploration of the MSE estimate is left for future work.
>
> - Soleymani et al. "Causal Feature Selection via Orthogonal Search." 2022
>
> > I note that the simulation data is only generated by a linear model. It should be clearly defined if the proposed theoretical results only focus on the linear model.
>
> We are sorry that this is not made clear in the paper. The proposed theoretical results are not dependent on linear data generation. For problems where data generation is non-linear, a non-linear invariance test can be swapped out for the linear invariance test used in the experiment. For example, see Heinze-Demel et al. 2018.
>
> - Heinze-Deml et al. "Invariant causal prediction for nonlinear models." 2018
>
> > If the non-parents node has no noise term, e.g., X_2=2E and X_1=X_2, it seems that the MMSE(X_1 X_2) is equal to (X_1), which may not ensure the correct ICP set output by the proposed method. In other words, the author should claim the causal model used in this paper, such as SCM or ANM. If my understanding is not correct, can you clarify what generation process assumptions you are making?
>
> We thank the reviewer for pointing out this case.
> In the noiseless case, it is true that MMSE(X_1, X_2) == MMSE (X_1). MMSE-ICP will not pick {X_1, X_2} over {X_1}. This is because MMSE-ICP will first test for {X_1} then test for {X_1, X_2} (line 3, Algorithm 1). When MMSE-ICP finds that {X_1} is invariant, it will skip over {X_1, X_2} (line 4, Algorithm 1).
>
> However, it may be possible that MMSE(X_1) == MMSE(X_2) if X_1 is a deterministic function of X_2. Thus, our approach may output {X_2} instead of {X_1}. Thus, it is true that our approach depends on the assumption that there is noise in the system. However, it does not assume that the noise is additive (ANM) or the generation follows some specific structural causal model (SCM).
>
> > My question is how to test the constraint "isInvariant(Y,S)" (refer to the algorithm). Can you give some illustrations to its implementations?
>
> The 'IsInvariance' operation is the same as used in prior work (Peters et al., 2016) and (Mogensen et al., 2022). Specifically, it tests whether the means and variances of the average prediction errors (residuals) across all environments are the same. The pseudo code is given below.
>
> ```
> p_vals = []
> for E_label in groups:
> 		in_group, out_group = residuals[E == E_label], residuals[E != E_label]
> 		p_val1 = t_test(in_group, out_group) // returns p-value
> 		p_val2 = levene_test(in_group, out_group) // returns p-value based on (Levene 1960)
> 		p_vals.append(2*min(p_val1, p_val2))
> return min(p_vals) < threshold
> ```
>
> - Levene. "Robust tests for equality of variances". 1960

---

> > ### Comment · Reviewer_ZvvZ · 2023-11-18
> > **Thank you for the response**
> >
> > Thank you for the response. I keep my score due to (i). the estimation of MSE may rely on some trick, and It is difficult to guarantee its convergence under the non-linear function model theoretically; (ii) the problem statement requires further clarity, such as the applicable model and its discussion.

---

> ### Author Response · Authors · 2023-11-20
>
> We thank the reviewer for the prompt response.
> We would like to clarify the two points raised.
>
> (i) As explained in Section 4.2, our estimate of the MSE is a straightforward average of the predicted residuals across environments.
> It is obtained by (1) fitting a linear model to predict Y using the data from all environments, (2) calculating the prediction error for each sample, (3) average the prediction errors.
>
> We are unsure what "convergence" means in the context of causal discovery.
>
> As for whether the MSE estimation is good enough for problems with non-linear data-generation, our SOTA result on the real data benchmark of Kemmeren et al. (2014) demonstrates, our proposed approach can be useful for practical applications. In addition, we want to emphasize that improving the MSE estimation is orthogonal to the main contributions of this paper. Our proposed approaches will only get better with more accurate MSE estimation. Finally, we would like to underscore that under the infinite (large-enough) data regime, an adoption of an infinite (large-enough) capacity (non-linear) model, instead of a linear model, will theoretically guarantee that the proposed approach can achieve an unbiased (arbitrarily good) estimate of minimum MSE.
>
>
> (ii) We are not sure what is meant by "applicable model" and would be grateful for further explanation.
> We believe our statement of the problem we aim to tackle has been sufficiently clear.
> As we describe in the introduction, our main goal is to identify the causal parents of a target variable. The causal parents can be used to build robust ML models or to design interventions to change the target variable value.
> As we describe in Section 3, we rely on the error inequality to implement an efficient version of ICP.

---

> ### Comment · Reviewer_ZvvZ · 2023-11-21
> **Thank you for the response**
>
> Thank you for the response. For the first question, I mean that whether a non-linear invariance test is fragile if data generation is non-linear, eg., fitting a non-linear model.
>
> Secondly, "applicable model"  means "assumptions for the data generation".
>
>
> There are some suggestions:
>
> (1). provide a more detailed definition to "environments", and illustrate "how to fit a linear model to predict Y using the data from all environments".
> (2). formalize the data generation process, which may be an important "problem definition" in causal discovery.
> (3). distinguish/explain the "intervention" and "environment" if possible.
>
>
> Overall, based on the author's response, most of my issues are addressed. I made some slight adjustments to the scores, and I suggest that the feedback should be incorporated into the paper.

---

> ### Author Response · Authors · 2023-11-21
>
> We thank the reviewer for the clarification and constructive feedback.
>
> We assume that the data are generated following an underlying causal system that can be represented with a DAG, in which the value of each variable/node is generated via a causal **mechanism** based on the values of its parents.
> In the DAG, there is a target variable Y and all other variables are considered as Xs.
> Some Xs may be ancestors of Y, while some others may be descendants.
>
> Following common terminology, we use **intervention** to refer to a change of **mechanism** at a variable.
> The data from an **environment** are generated from the same set of **mechanisms**.
> Two different **environments** can have some differing **mechanisms**.
> However, the **mechanism** at Y, i.e., the way Y is generated from its parents, is assumed to be unchanged (or invariant) across all **environments**.
> We assume that the knowledge of where and how the remaining **mechanisms** differ between two **environments** are unknown.
> Since the location of the **mechanism** changes are not known, these are often referred to as **unknown-target interventions**.
> All of these assumptions are identical to the setup in Peters et al. and Mogensen et al.
>
> - Peters et al. "Causal inference by using invariant prediction: identification and confidence intervals." 2016
> - Mogensen et al. "Invariant Ancestry Search." 2022
>
> Although these assumptions are standard in the invariant causal prediction literature, we realized that the wider causal inference community may be unfamiliar with this setup, especially the concept of **unknown-target interventions**.
>
> We will update the draft to integrate all the comments above to make the problem setup clearer.

---

### Official Review · Reviewer_DSFM · 2023-10-31

**Soundness:** 3 good
**Presentation:** 2 fair
**Contribution:** 2 fair
**Rating:** 6
**Confidence:** 4

**Summary:**

This paper proposed to identify the invariance subset of covariates across different domains, by exploiting the property of mean squared error (MSE). Compared to the original ICP, the proposed method required fewer interventions in identification. Further, a more scalable version called FastICP was proposed. The complexity analysis was provided. Experiments on conducted on synthetic datasets and real-world datasets.

**Strengths:**

1. Theoretical analysis regarding the identification of the invariance set and the complexity analysis are provided.
2. Extensive experiments are conducted. Particularly, the application on a large graph demonstrates the effectiveness of the proposed methods.
3. Generally, this paper is well-organized.

**Weaknesses:**

1. It would be practically important to identify the set $PA(Y)$ compared to $DE(E) \cap PA(Y)$, which may be done by exploiting MSE since the former has smaller MSE if $PA(Y) \cap (DE(E))^c \neq \emptyset$.
2. The specific procedures for the 'IsInvariance' operation (located at line 1 in Algorithm 1) are not provided, although it appears to involve a test for conditional independence.
3. The running time should be provided on synthetic dataset.

**Questions:**

Please see the weakness above.

**Details Of Ethics Concerns:**

Not applicable.

---

> ### Author Response · Authors · 2023-11-14
>
> > It would be practically important to identify the set $PA(Y)$ compared to $DE(E)\cap PA(Y)$, which may be done by exploiting MSE since the former has smaller MSE if $PA(Y)\cap(DE(E))^c≠\emptyset$.
>
> It is true that $PA(Y)$ has smaller MSE than $DE(E)\cap PA(Y)$, when the latter is a strict subset of the former. However, it may not be possible to identify $PA(Y)$ since we do not assume that the data contain interventions at every node $X$.
>
> When the data contain interventions at every node $X$, $PA(Y)$ is the only set that achieves “invariant” prediction error across environments/domains/sites. Unfortunately, due to logistical constraints, the data gathered may not contain interventions at every node. For example, when a node $X$ denotes weather and it was always sunny during the data gathering period, then no intervention on $X$ has been observed in the data.
>
> When the number of observed interventions in the data is fewer than the number of $X$s, there are many sets that can achieve “invariant” prediction error. These sets are deemed “invariant” only because of the limitation of the data gathering. If we have seen more interventions, we could have rejected all the other sets that are not $PA(Y)$.
>
> Consider the following 3 “invariant” sets:
>
> - $DE(E) \cap PA(Y)$
> - $PA(Y)$
> - $PA(Y) \cup (CH(Y) \cap (DE(E))^c)$
>
> They are in increasing order of cardinality and in decreasing order of MSE. It is not possible to pick out $PA(Y)$ because $PA(Y)$ has neither the smallest cardinality nor the smallest MSE among all the “invariant” sets. If we chose the “invariant” set with minimum MSE, then we would have chosen a set that includes not only parents of $Y$ but also children of $Y$.
>
> Instead of considering all “invariant” sets, MMSE-ICP and fastICP only consider “invariant” sets that are subsets of $DE(E) \cap ND(Y)$. Among the subsets of $DE(E) \cap ND(Y)$, it is possible to identify $DE(E) \cap PA(Y)$ because it has the minimum MSE.
>
> > The specific procedures for the 'IsInvariance' operation (located at line 1 in Algorithm 1) are not provided, although it appears to involve a test for conditional independence.
>
> The 'IsInvariance' operation is the same as used in prior work (Peters et al., 2016) and (Mogensen et al., 2022). Specifically, it tests whether the means and variances of the average prediction errors (residuals) across all environments are the same.
>
> - Peters et al. "Causal inference by using invariant prediction: identification and confidence intervals." 2016
> - Mogensen et al. "Invariant Ancestry Search." 2022
>
> > The running time should be provided on synthetic dataset.
>
> We recorded the time each method took when the number of nodes and the number of interventions are equal to 6 (this corresponds to the scenario in Fig 2). This scenario is chosen because all the methods can complete exhaustive search for a problem of this size, thus allowing us to conduct a numerical comparison between all methods. For problems with 100 variables, ICP and IAS will not be able to search through all subsets.
>
> The table below reports the numbers of seconds elapsed on average when executing on an Intel Xeon 6126 CPU core (@ 2.60GHz).
> N denotes the sample size.
> The numbers in brackets are the standard deviation across different simulations.
>
> Since the official implementation of FGES_MB, ICP, and ICP_CORTH are not written in Python so direct comparison with fastICP should be viewed with caution.
> However, we can see that:
>
> 1. Among ICP-like methods with exponential worst-case complexity, fastICP is faster than MMSE-ICP, ICP, ICP_CORTH, and IAS.
> 2. fastICP is also slightly faster than UT-IGSP for large sample size (1E5 samples).
>
> | Method      | Language  |     N=1E2     |     N=1E3    |     N=1E4   |      N=1E5  |
> |-------------|-----------|---------------|---------------|---------------|----------------|
> | FGES_MB     | Java      | 2.342 (0.028) | 0.246 (0.004) | 0.364 (0.006) |  1.340 (0.019) |
> | UT-IGSP     | Python    | 0.026 (0.002) | 0.040 (0.008) | 0.059 (0.017) |  0.607 (0.223) |
> | ICP         | R         | 0.190 (0.006) | 0.325 (0.004) | 1.436 (0.026) | 20.296 (0.421) |
> | ICP_CORTH   | R         | 1.662 (0.104) | 1.857 (0.122) | 3.759 (0.543) | 28.312 (7.061) |
> | IAS         | Python    | 0.016 (0.007) | 0.056 (0.010) | 0.112 (0.022) |  0.659 (0.132) |
> | MMSE-ICP    | Python    | 0.035 (0.010) | 0.067 (0.015) | 0.138 (0.027) |  0.932 (0.187) |
> | fastICP     | Python    | 0.028 (0.002) | 0.036 (0.002) | 0.070 (0.004) |  0.516 (0.030) |
>
> This table has been added to Appendix C.

---

### Official Review · Reviewer_ru9u · 2023-11-04

**Soundness:** 3 good
**Presentation:** 3 good
**Contribution:** 3 good
**Rating:** 6
**Confidence:** 3

**Summary:**

This paper tackles the limitations of existing methods (e.g., ICP) that identify causal parents of a target via exploiting distribution shifts. Two approaches are proposed based on an error inequality and have good theoretic guarantees of identifiability of invariant variables. Experiments also show imporved results.

**Strengths:**

- Two novel algorithms for finding causal parents of a targets, with both theoretic and empirical validations.

- Presentation is really good.

**Weaknesses:**

Testing invariance may not be easy for general problems.

**Questions:**

The paper writing is clear and I have a few questions/suggestions:

- It is claimed that the number of interventions required can be much less than the number of variables in the graph, which improves ICP. In my understanding, this is because both observed information (utilized by traditional causal discovery method like GES and PC) and interventional information are used. Is this correct? Please highlight and discuss this point more in the paper.

- in the experiment, fig. 5, ICP and the proposed methods have close F1, but ICP's recall is much lower. Does ICP have a higher precision in this case?
- please use \citet and \citep separately. This really affects reading flow.

Overall, I think this paper improves an existing method with both theoretic and empirical validations. The current version is in a good shape and only needs minor edits. I vote for an acceptance.

---

> ### Author Response · Authors · 2023-11-14
>
> > Testing invariance may not be easy for general problems.
>
> It is true that leveraging invariance testing to build generalizable ML models in general problems has not been widely explored. However, this is partly due to the limitations of ICP since ICP (1) cannot tackle problems with many variables and (2) does not give informative outputs when there are insufficient interventions. These limitations have been explored in (Arjovsky et al. 2019); (Rosenfeld et al. 2021); and (Mogensen et al., 2022). The proposed fastICP addresses both of these limitations and therefore promises to expand the applicability of invariance testing.
>
> In addition, our proposed approach does not require knowing which variables are intervened upon (unknown intervention targets) so it can be easily applied to problems where the data come from multiple domains/sites.
>
> - Arjovsky et al. "Invariant risk minimization." 2019
> - Rosenfeld et al. "The Risks of Invariant Risk Minimization." 2021
> - Mogensen et al. "Invariant Ancestry Search." 2022
>
> > It is claimed that the number of interventions required can be much less than the number of variables in the graph, which improves ICP. In my understanding, this is because both observed information (utilized by traditional causal discovery methods like GES and PC) and interventional information are used. Is this correct? Please highlight and discuss this point more in the paper.
>
> ICP, MMSE-ICP, and fastICP all use observational and interventional data.
> However, our methods differ from ICP in the way they pick out the set of causal parents from multiple detected invariant sets.
>
> - ICP takes the intersection of the invariant sets. When there are insufficient interventions, there can be multiple non-overlapping invariant sets so taking the intersection results in a non-informative output.
> - Our methods select the best set from the given invariant sets by leveraging the error inequality (Theorem 1). As our experiments demonstrate, this usually yields more informative outputs than ICP.
>
> > In the experiment, fig. 5, ICP and the proposed methods have close F1, but ICP's recall is much lower. Does ICP have a higher precision in this case?
>
> For the result in Fig. 5, ICP only has higher precision when the number of samples is 1E5.
>
> > Please use \citet and \citep separately. This really affects reading flow.
>
> We thank the reviewer for the suggestion. We have updated the paper accordingly.
>
> > Overall, I think this paper improves an existing method with both theoretical and empirical validations.
>
> We thank the reviewer for the positive feedback.

---

> > ### Comment · Reviewer_ru9u · 2023-11-18
> >
> > Thanks for clarifications and I'm happy to see this paper be accepted.

---

> ### Author Response · Authors · 2023-11-20
>
> We thank the reviewer for the prompt response.

---

### Author Response · Authors · 2023-11-14

We thank the reviewers for their constructive and insightful feedback. We provide a point-by-point response to each reviewer’s comment/questions below their corresponding review.

---

### Meta-Review · Area_Chair_9RB5 · 2023-12-15

**Metareview:**

This paper introduces methods that improve upon Invariant Causal Prediction (ICP) by using an error inequality principle to identify causal parents of a target variable efficiently. Despite its theoretical foundation and empirical validation, the paper falls short somewhat in detailed methodology, particularly in "convergence" in the context of causal discovery and clarifying causal model assumptions.

**Justification For Why Not Higher Score:**

The paper has several drawbacks.

**Justification For Why Not Lower Score:**

NA

---

### Decision · Program_Chairs · 2024-01-16

Reject